# Association of Iron Storage Markers with Metabolic Syndrome and Its Components in Chinese Rural 6–12 Years Old Children: The 2010–2012 China National Nutrition and Health Survey

**DOI:** 10.3390/nu12051486

**Published:** 2020-05-20

**Authors:** Huidi Zhang, Lijuan Wang, Siran Li, Xiaobing Liu, Yuqian Li, Yuna He, Qingqing Man, Lichen Yang

**Affiliations:** National Institute for Nutrition and Health, Chinese Center for Disease Control and Prevention, Beijing 100050, China; zhanghuidi1114@126.com (H.Z.); Wanglj@ninh.chinacdc.cn (L.W.); sirancara@163.com (S.L.); liuxb@ninh.chinacdc.cn (X.L.); liyq@ninh.chinacdc.cn (Y.L.); heyn@ninh.chinacdc.cn (Y.H.); manqq@ninh.chinacdc.cn (Q.M.)

**Keywords:** MetS components, whole blood iron, ferritin, total body iron, Chinese children

## Abstract

Background: Elevated ferritin, which is often used to represent iron storage, is known to increase the risk of metabolic syndrome (MetS) or its components, but its increase is affected by many factors. Therefore, it is necessary to analyze the relationship between other indicators of iron storage, and MetS and its components in order to fully understand the role of iron in the occurrence and development of these diseases. Although there are many studies to analyze the relationship involved in adults and adolescents, in children there is limited research. In this study, we aim to estimate the association of whole blood iron, ferritin, and total body iron with metabolic syndrome, and especially its components in Chinese rural children aged 6–12 years old. Method: A total of 1333 children aged 6–12 years old were enrolled from the 2010–2012 China National Nutrition and Health Survey in this study. Markers of iron storage (whole blood iron, ferritin, and total body iron (TBI)) and MetS component parameters (waist, blood pressure, high-density lipoprotein cholesterol (HDL-C), triglyceride (TG), and fast glycose) were collected. A multivariate logistic regression analysis was performed to confirm the independent relationship between iron storage markers, and the incident of metabolic syndrome and its components. Results: After adjusting for age, gender, C-reactive protein (CRP), and body mass index (BMI), a negative association was found between whole blood iron, ferritin, and TBI and incidence of reduced HDL-C (odds ratio (OR) = 0.63, 0.49, and 0.57, respectively). The highest tertile of whole blood iron increased the risk of the incidence of hyperglycemia (OR = 1.74), while TBI decreased the risk by 61%. No significant association was found between ferritin tertiles and the incidence of hyperglycemia. Conclusion: An iron storage level within the normal range in children is associated with a risk of MetS components, especially in hyperglycemia and reduced HDL-C. The relationship between the three iron indexes and metabolic syndrome and its components is not completely consistent, which suggests that the underlying mechanism is complex and needs to be further explored.

## 1. Introduction

During the past decades, metabolic syndrome (MetS) has become a major public health issue in the world [1]. It is a predictor of cardiovascular diseases (CVDs) and type 2 diabetes mellitus [2,3]. With the worldwide prevalence of obesity in children, the incidence of metabolic syndrome in children and adolescents has gradually increased [4]. The prevalence of MetS from the 2010–2012 Chinese National Nutrition and Health Survey (CNNS) was 2.4% in children aged 10 to 17 years old, and presented a low age trend [5].

Iron is an important trace element for humans to sustain the function of development [6]. The hypothesis that iron status could influence the risk of coronary heart disease was first proposed by Sullivan in the 1980s [7]. Since then MetS and its components have received widespread attention and discussion. Recent studies have also shown that rising ferritin will increase the risk of MetS or its components, but the specific mechanism of this is still unclear [8,9]. Ferritin often represents the iron storage level of the body, but its increase is affected by many factors, such as inflammation and injury. Therefore, it is necessary to analyze the relationship between other indicators of iron storage and MetS and its components in order to fully understand the role of iron in the occurrence and development of these diseases. Serum ferritin [10], serum transferrin receptor (sTfR) [11], the sTfR–ferritin ratio [12], and dietary iron [13] are often used as the common measurement in the assessment of body iron storage. The function of whole blood iron and total body iron concentrations have yet to be explored as an indicator of iron nutrition status. The relationship between iron storage and MetS in adults has been basically demonstrated in [14,15,16]. There are also some studies that analyze this relationship in adolescents [17]. However, when it comes to a much younger group, there are few studies. 

Thus, in this research, we aimed to assess the correlation between markers of iron storage and MetS and, in particular, its components in Chinese children.

## 2. Materials and Methods

### 2.1. Subjects 

The study was based on the data obtained from the 2010–2012 China National Nutrition and Health Survey (CNNS), which is a cross-sectional, nationally representative survey. We randomly selected 1400 participants from the 2010–2012 CNNS aged 6–12 years old, in a rural area, with complete physical detection. The subjects were sampled by a multistage stratified random sampling method. All of the subjects gave their informed consent for inclusion before they participated in the study. The study was conducted in accordance with the Declaration of Helsinki, and the protocol was approved by the Ethics Committee of the Institute for Nutrition and Food Safety, Chinese Center for Disease Control and Prevention (now changed to the National Institute for Nutrition and Health, China CDC; file number 2013-018).

### 2.2. Data Collection

Physical examinations were performed by trained medical staff following standardized procedures. Height was measured by a Seca 213 Portable Stadiometer Height-Rod with a precision of 0.1 cm, and body weight was measured by a Seca 877 electronic flat scale with a precision of 0.1 kg. Body mass index (BMI) was calculated as weight (kg)/square of height (m^2^). Waist circumference was measured by a tape with a precision of 0.1 cm. Systolic blood pressure (SBP; mmHg) and diastolic blood pressure (DBP; mmHg) were assessed twice in the right upper arm using an Omron HBP-1300 professional BP monitor. The second of the two measurements was recorded if the difference between the two was less than 10 mmHg. A third measurement was performed if the difference between the first two measurements was more than 10 mmHg, and any two measurements with a difference of less than 10 mmHg were recorded. The venous blood was collected and divided into the anticoagulation tube and serum separator tube, separately. The blood samples in the serum separator tube were promptly centrifuged at 3000× *g* for 15 min after blood collection, divided into an aliquot of serum, and frozen at −80 °C for subsequent assays. Serum fasting glucose, high-density lipoprotein cholesterol (HDL-C), triglyceride (TG), C-reactive protein (CRP), serum ferritin, and sTfR concentrations were measured by an enzymatic method using Hitachi 7600 automatic biochemical analyzer (Japan). The whole blood Fe concentration was measured by inductively coupled plasma mass spectrometry (ICP-MS) from the anticoagulation tube. The concentration of total body iron is measured using ferritin and sTfR, and it is calculated as the following formula: total body iron (TBI) = −[log(sTfR/SF) − 2.28229]/0.1207 [18].

### 2.3. Definitions of MetS and Its Components

As the subjects in this study were aged between 6 and 12 years old, no consistent definition of metabolic syndrome (MetS) could be used here. We adopted the modified criteria for MetS from the American Academy of Pediatrics (AAP) and the International Diabetes Federation (IDF) [19], which indicate MetS as the presence of ≥3 of the risk factors. The MetS components were followed as: (1) obesity: waist ≥95th percentile of children of the same age and gender, or BMI ≥95th percentile of children of the same age and gender; (2) hypertension: blood pressure ≥95th percentile of children of the same age and gender (fast identified: systolic BP ≥120 mmHg or diastolic BP ≥80 mmHg); (3) dyslipidemia: (a) reduced HDL-C (<1.03 mmol/L) or (b) elevated TG (≥1.47 mmol/L); and (4) hyperglycemia: fasting glucose ≥5.6 mmol/L.

### 2.4. Statistical Analysis

All of the samples were re-examined after the physical and biological data collection, with the following exclusion criteria: (1) iron deficiency (defined as a ferritin level of <12 µg/L [20]); as the first tertile was used as the reference in the logistic regression and it was also the lowest tertile, we needed to eliminate the confusion caused by iron deficiency, which could cause a mixed bias and reduce the sensitivity of the whole study; (2) missing data for variables; (3) data do not meet test standards; and (4) outliers for statistical analysis. because of the exclusion of some samples, 1333 samples were finally recruited in this study. The whole blood iron (*n* = 911), ferritin *(n =* 521), and total body iron (TBI; *n* = 521) were used as the markers of iron metabolism. Statistical analyses were performed using SPSS version 19.0. The results of the descriptive characteristics were expressed as the average ± standard deviation. The relationships among the clinical indexes and the number of MetS components were analyzed using the Kruskal–Wallis test. The odds ratios (ORs) and 95% confidence intervals (95% CI) were determined by multivariate logistic regression so as to investigate the associations between MetS and its components, and the tertiles of the markers of iron metabolism. We categorized the level of whole blood iron, ferritin, sTfR, and TBI into tertiles (whole blood iron (mg/L): <401.7, 401.7–439.4, >439.4; ferritin (µg/L): <50.0, 50.0–88.6, >88.6; and TBI (mg/kg): <3.42, 3.42–5.72, >5.72), using the lowest tertile as the reference. A *p*-trend analysis was done by treating the tertiles as a continuous variable in the regression analyses. All of the statistical tests were two-sided, and statistical significance was determined as *p* < 0.05.

## 3. Results

### 3.1. Characteristics of the Study Population

Demographic and clinical data of the participants are shown in Table 1. Among the 1333 subjects, 675 were male and 658 were female. When comparing the baseline characteristics between the different genders, the value of BMI (16.59 ± 2.89 vs. 16.14 ± 2.54 kg/m^2^), waist (56.24 ± 7.77 vs. 54.80 ± 7.64 cm), HDL-C (1.29 ± 0.29 vs. 1.21 ± 0.28 mmol/L), FG (4.89 ± 0.73 vs. 4.78 ± 0.65 mmol/L), and sTfR (3.91 ± 1.32 vs. 3.59 ± 0.91 mg/L) was much higher in the males than in the females (*p* < 0.05), while TG was inversely lower in the males than in the females (0.74 ± 0.52 vs. 0.83 ± 0.48 mmol/L). As the number of MetS components increased, there was a significantly positive association observed in height, weight, BMI, waist, TG, FG, SBP, and DBP (*p* < 0.05 for trend). A significant inverse association was noted between the HDL-C and MetS components. Individuals with two or more MetS components had much higher obesity rates than the other groups. Although, with the increase of metabolic syndrome components, the trend of the iron storage indexes was not statistically significant; the concentration of ferritin, whole blood iron, and sTfR was still the highest in the more than three MetS component groups. TBI was all positive in all of the four MetS components groups. 

### 3.2. The Association between Tertiles of Markers of Iron Storage and MetS Components Indicators

Table 2 shows the characteristics of the study subjects according to the tertiles of whole blood iron, serum ferritin, and TBI concentrations. BMI, waist, HDL-C, FG, SBP, and DBP concentrations were positively associated with the whole blood iron tertiles (*p* < 0.05). In the analysis of genders, the HDL-C concentrations were significantly higher with whole blood iron in tertiles in the males, but not in the females. The increase in SBP was consistent with the iron tertile only in the females. BMI, FG, and DBP were also associated with iron in either men or women. 

In the assessment of the relationship between ferritin tertiles and MetS component indicators, a positive association only existed in HDL-C and DBP. There was a significant difference in SBP concentrations in the males and HDL-C concentrations in the females as the tertile increased. 

In the TBI tertiles, the concentrations of HDL-C increased significantly in all of the participants. There were no associations between TBI and the other indicators. The consistency of the three markers of iron storage related to the MetS components existed in HDL-C, while the DBP concentrations increased with the ferritin and whole blood iron in the tertiles.

### 3.3. Odds Ratios for MetS and Its Components in Tertiles of Markers of Iron Storage

The results of the multivariate logistic analysis of the associations between tertiles of the markers of iron status, and incidents of MetS and its components are summarized in Table 3. It demonstrates the risks of MetS components according to the tertiles of whole blood iron, ferritin concentration, and TBI before and after adjustment.

There was a positive association between the tertiles of whole blood iron and incidents of hypertension and hyperglycemia, and the odds ratios (ORs) for individuals categorized in tertile three after adjusting for age, gender, CRP, and BMI was 4.07 and 1.74, respectively. The incidence of reduced HDL-C was also significantly associated with whole blood iron tertiles, but it showed a negative association (OR = 0.63) in the adjustment model. Models evaluating the trend of the above associations were also statistically significant (*p* < 0.05). No significant associations were observed in the elevated waist or in the elevated TG. In the ferritin tertiles, the decreased risk for reduced HDL-C of the third ferritin tertile compared with the first tertile for all of the participants was 51%. Higher ferritin levels were not associated with higher odds of other MetS components, even after adjusting confounding factors. In the adjusted model in TBI, the higher tertiles decreased the risk of incident for elevated TG (OR = 0.31), reduced HDL-C (OR = 0.57), and hyperglycemia (OR = 0.39). There was no significant association between TBI tertiles and incident of hypertension and elevated waist. For all iron indices, the consistency of investigating the risk of MetS and its components was found to be in elevated HDL-C. They all could decrease the prevalence risk of reduced HDL-C.

## 4. Discussion 

MetS and its components have received widespread attention and discussion since MetS was raised in 1981 by Sullivan [7]. Iron overload is well known to cause chronic noncommunicable diseases in individuals with obesity [21]. It is of interest to investigate the effect of iron metabolism in the normal range on the development of MetS. The purpose of our study is to explore whether SF, whole blood iron, and TBI will increase the risk of MetS or its components in the high tertile of the normal distribution by using a representative population. In this cross-sectional study, we initially observed that there was no correlation between tertiles of iron metabolic markers and MetS, but there were associations between iron metabolic markers and MetS components. Stronger associations were observed between iron markers and reduced HDL-C or high fasting glucose than with other components of MetS. We found that high levels of whole blood iron, ferritin, and TBI were negatively associated with the incidence of reduced HDL-C. In addition, higher levels of whole blood iron were positively associated with hyperglycemia, while inversely negatively associated with TBI. From the 1970s to the present, ferritin has been regarded as a clinical indicator of iron storage [21]. The threshold of SF recommended by the World Health Organization (WHO) for determining iron excess is 200 µg/L for women and 300 µg/L for men. However, the cut-off value is only for adults and there is no consistent recommendation for children. It was also found that a high SF level can increase the risk of MetS when the threshold of iron excess is not reached [8,22]. However, based on the current standard, the values of SF in our study population were all within the normal range. SF is also an acute-phase protein, which could be raised under inflammation and injury. As a consequence, we adjusted CRP as a confounding factor in order to eliminate inflammatory effects on the serum ferritin and MetS components. Iron overload is well known to cause chronic noncommunicable diseases in individuals with obesity [23]. In the study of Matthew [24], it is indicated that iron overload can lead to lipid peroxidation and further result in obesity, while obesity can cause chronic inflammation, which can lead to a change in iron storage. In the meanwhile, studies in China [25], Korea [26], and Switzerland [27] also show that BMI was found to influence the ferritin–MetS association, and was used as a confounding factor. In this study, we also made the same attempt. BMI was adjusted to assess the pure effects. In order to support the idea that iron storage may be associated with MetS, we needed to combine more iron status markers for verification. Based on the 2010–2012 blood sample restriction, two other markers, whole blood iron and TBI, which are not commonly used together with serum ferritin, but are vital in defining the iron status, were also analyzed in our research. TBI, which combines ferritin and sTfR, is proven to be a reliable assessment of body iron status, which is superior to the measurement of serum ferritin and sTfR alone in epidemiological settings [28]. It can rectify the single factor, which is easily influenced by many confounding factors, and can also replace the bone marrow iron staining with more easily and accurately [29]. In the American National Health and Nutrition Examination Survey 2003–2004, a TBI model was used to assess iron status in the replacement of ferritin [29]. To date, there are few studies that have used TBI as an indicator of iron storage and that also detects the relationship between TBI and MetS components. As for whole blood iron, there are also few reports using whole blood iron as an indicator of iron status, but it can also reflect the nutritional status of iron, and can be quickly detected clinically with multiple elements at the same time. However, there has been no comprehensive evaluation of the interaction between whole blood iron and metabolic syndrome. Whether it is whole blood iron or TBI, both are simple and easy-to-measure indicators, especially for screening for the iron nutrition status in a large population. It is worth finding a simpler approach involving fewer iron status indicators, which may also reduce survey complexity and cost.

Although no association was found between the incident of MetS and iron status markers in this study, contrary to other studies [30], it could be explained by the small sample size of the confirmed cases *(n =* 13) and because no consensus has been reached for the definition MetS in such a young age group [31,32]. Another possible explanation would be that most children in this study had only one or two MetS components, which were insufficient to be diagnosed with MetS. Therefore, this article focuses more on the relationship between iron storage indexes and metabolic syndrome components. Most of the metabolic syndrome components were concentrated in reduced HDL-C and hyperglycemia.

The main possible explanations of the correlation between the iron storage markers and glucose and lipid metabolism include the factors of oxidative stress, inflammation, and insulin resistance [33,34]. When excess iron accumulates in the muscle and adipose tissue, it will accelerate lipolysis and decrease glucose oxidation because of tissue damage [35,36]. Tissue damage also includes the blood vessels, which could lead to hypertension and vascular injury. Thus, this also explains the result in our study, that high whole blood iron tertiles increased the risk of incident for hypertension. Furthermore, when a higher level of iron conducts a high oxidation activity in the pancreas, it will damage the function of the β-cell and finally lead to insufficient insulin production [37,38,39]. Insulin is an important regulator of glucose metabolism. Insulin damage would lead to the imbalance of blood glucose regulation, which would be a possible explanation for the higher whole blood iron associated with an increased risk of hyperglycemia incidence, but does not explain the result of TBI, which still needs further exploration in order to verify the unexpected result.

Unexpectedly, in this study, the whole blood iron, serum ferritin, and TBI concentrations showed a negative association with reduced HDL-C. This is contrary to previous research in adults and the possible mechanisms. However, in the assessment of children, this study is not the only conflicting one. A study conducted in France in 8–15-year-old children to assess iron metabolism and its association with dyslipidemia risk also showed a significant negative correlation in reduced HDL-C and ferritin levels (the concentration of ferritin in the study was 41.7 ± 22.9 µg/L) [40]. A study of Korean adolescents also found that serum ferritin levels ((the concentration of ferritin in the study was 42.7 (40.7,44.9) µg/L) were negatively associated with HDL-C values, even after adjusting for all covariates [41]. One possible reason is that, unlike other reports [17,42,43], the distribution of iron storage indicators in these studies is within the normal range. The definition of iron surplus of TBI is beyond zero [29]. There is no cut-off value of whole blood iron to define the status of iron overload in children. Ferritin concentrations are commonly considered to be normal within the range of 15–300 µg/L [44]. No matter the concentration of ferritin in this study (78.31 ± 49.37 µg/L) or the studies mentioned before, none would be considered as iron excess, based on the criteria commonly used now. The differences may also partially be explained by the response of ferritin towards inflammation, which was quite different in the subjects with and without dyslipidemia in this study. As a consequence, whether higher iron storage will lead to reduced HDL-C, and the function of this result, still needs further verification.

There are some advantages to this study. First of all, we did not only use ferritin as the indicator of iron status, as its shortages are easily affected by inflammation, but also used whole blood iron and TBI. The combination of these three markers could more effectively reflect the storage of iron. This is the first time these three markers have been combined with MetS components in Chinese children. It could provide a new idea and evidence to assess the effect of iron storage on MetS components. The second advantage is the population we evaluated, 6–12-year-old children in rural China. The NCEP ATP III criteria [45] only applies to children above 16 years of age, but are not yet particularly applicable to younger children. Therefore, the relationship between iron status and cardiovascular disease risk factors in children is difficult to accurately assess. It is noteworthy that this also provided unique evidence to study the association between the markers of iron metabolism and MetS components, especially in Chinese children aged 6–12 years old.

The potential limitations of this study are as follows. Firstly, we did not control insulin resistance in the analysis, as it was not tested in this study, and it would lead to an underestimation of the effects of iron metabolism on dyslipidemia and hyperglycemia. Second, CRP is one of the indexes of inflammation, which has a very rapid response to inflammation. Although the CRPs in our study were all in the normal range of 8–80 mg/L, the effect of using CRP alone for correcting for all inflammation is very limited. It cannot remove all inflammatory effects. Therefore, more inflammatory markers need to be measured, such as α-1-acid glycoprotein (AGP), α-1-antichymotrypsin (ACT), and fibrinogen. Third, the sample size for a large national survey is too small to explain all of the assessments in this study, but it can be representative of rural Chinese children. The fourth is that MetS is highly unstable throughout childhood, as multiple observational longitudinal studies have shown [46,47]. However, it is still worth estimating, as children are at an important stage of growth and development. Their current physical condition has an important impact on their future. As a consequence, the association of iron storage and MetS needs a long-time follow-up in order to be fully assessed. As we know, there are some reports about the relationship between the intake of total iron, especially heme iron in the diet, and MetS [48,49], but mostly for adults. In this study, three biological indicators reflecting iron storage were used to evaluate the relationship between body iron status and MetS. However, we did not collect dietary data, especially heme iron data. More rigorous and evidence-based epidemiological studies, such as cohort studies, are needed in order to make recommendations on iron consumption in the population.

In conclusion, the three markers (within the normal range) were all associated with a risk of MetS components and reduced HDL-C in Chinese children aged 6–12 years old. The high level of iron storage markers in this study was negatively associated with incidents of reduced HDL-C. The high levels of whole blood iron were positively associated with hyperglycemia, while TBI was inversely negatively associated. Further study could be done to explore the function of iron storage markers in reduced HDL-C, especially in children, as well as the assessment of the normal reference range for iron storage markers.

## Figures and Tables

**Table 1 nutrients-12-01486-t001:** Basic characteristics according to gender and the number of metabolic syndrome (MetS) components.

Indexes	Total (*n =* 1333)	Male (*n* = 675)	Female (*n* = 658)	MetS Components			
				0 (*n* = 725)	1 (*n* = 491)	2 (*n* = 104)	≥3 (*n* = 13)
Age (years)	9.08 ± 1.71	9.01 ± 1.68	9.15 ± 1.74	8.98 ± 1.69	9.17 ± 1.72	9.30 ± 1.78	9.03 ± 1.69
Height (m)	1.31 ± 0.11	1.31 ± 0.11	1.31 ± 0.12	1.30 ± 0.10	1.31 ± 0.12	1.32 ± 0.14	1.36 ± 0.12 ^#^
Weight (kg)	28.65 ± 8.63	28.96 ± 8.91	28.32 ± 8.32	27.21 ± 7.00	29.62 ± 9.36	33.21 ± 11.56	35.53 ± 13.46 ^#^
BMI (kg/m^2^)	16.36 ± 2.73	16.59 ± 2.89	16.14 ± 2.54 *	15.82 ± 2.32	16.69 ± 2.82	18.37 ± 3.49	18.49 ± 3.46 ^#^
Waist (cm)	55.53 ± 7.74	56.24 ± 7.77	54.80 ± 7.64 *	53.38 ± 5.18	56.93 ± 8.61	62.85 ± 10.56	63.57 ± 12.53 ^#^
Obesity (%)	15.37%	13.18%	17.62%	0%	26.47%	64.42%	61.53%
TG (mmol/L)	0.78 ± 0.51	0.74 ± 0.52	0.83 ± 0.48 *	0.65 ± 0.28	0.89 ± 0.56	1.03 ± 0.62	1.91 ± 1.98 ^#^
HDL-C (mmol/L)	1.25 ± 0.29	1.29 ± 0.29	1.21 ± 0.28 *	1.37 ± 0.25	1.12 ± 0.27	1.00 ± 0.21	0.90 ± 0.25 ^#^
FG (mmol/L)	4.84 ± 0.69	4.89 ± 0.73	4.78 ± 0.65 *	4.70 ± 0.52	4.91 ± 0.73	5.21 ± 0.94	6.57 ± 1.23 ^#^
SBP (mmHg)	92.21 ± 11.53	92.71 ± 11.16	91.71 ± 11.88	91.36 ± 10.55	92.71 ± 12.23	94.82 ± 13.29	100.56 ± 15.40 ^#^
DBP (mmHg)	60.37 ± 8.81	60.60 ± 8.51	60.14 ± 9.11	59.66 ± 8.15	60.93 ± 9.36	62.32 ± 9.87	63.48 ± 10.54 ^#^
CRP (mg/L)	0.65 ± 0.83	0.65 ± 0.85	0.64 ± 0.81	0.62 ± 0.85	0.66 ± 0.83	0.66 ± 0.73	1.19 ± 0.99
Ferritin (µg/L)	78.31 ± 49.37	77.22 ± 49.95	79.43 ± 48.84	79.72 ± 49.19	76.69 ± 48.16	72.59 ± 46.78	103.25 ± 101.37
sTfR (mg/L)	3.75 ± 1.15	3.91 ± 1.32	3.59 ± 0.91 *	3.78 ± 1.21	3.63 ± 0.97	3.95 ± 1.18	4.75 ± 2.27
Iron (mg/L)	416.8 ± 52.92	416.4 ± 49.03	417.2 ± 56.77	415.7 ± 52.15	416.9 ± 52.63	422.9 ± 59.03	425.8 ± 62.68
TBI (mg/kg)	4.49 ± 2.25	4.31 ± 2.26	4.62 ± 2.23	4.53 ± 2.29	4.54 ± 2.20	4.08 ± 2.07	3.87 ± 3.61

MetS—metabolic syndrome; BMI—body mass index; TG—triglycerides; HDL-C—high-density lipoprotein cholesterol; FG—fasting glucose; SBP—systolic blood pressure; DBP—diastolic blood pressure; CRP—C-reactive protein; sTfR—soluble transferrin receptor; * *p* < 0.05 between male and female; # *p* < 0.05 for trend.

**Table 2 nutrients-12-01486-t002:** Distribution of MetS component indicators by tertiles of iron storage markers.

Indexes	Whole Blood Iron				Ferritin				TBI			
	T1	T2	T3	*p*-Value	T1	T2	T3	*p*-Value	T1	T2	T3	*p*-Value
**Total**												
BMI (kg/m^2^)	16.06	16.52	16.67	0.003	16.06	16.33	16.50	0.736	16.18	16.45	16.25	0.540
Waist (cm)	54.99	56.13	56.79	0.030	54.41	54.99	54.53	0.871	54.39	55.04	54.49	0.895
TG (mmol/L)	0.74	0.80	0.83	0.113	0.75	0.76	0.77	0.634	0.76	0.73	0.80	0.929
HDL-C (mmol/L)	1.23	1.28	1.30	0.016	1.18	1.19	1.26	0.010	1.19	1.21	1.24	0.033
FG (mmol/L)	4.76	4.80	5.00	<0.001	4.88	4.77	4.81	0.543	4.93	4.71	4.81	0.052
SBP (mmHg)	90.16	92.69	94.59	<0.001	90.25	91.69	92.38	0.058	90.22	92.37	91.72	0.212
DBP (mmHg)	58.63	60.50	61.66	<0.001	59.17	60.94	61.25	0.031	59.04	61.68	60.62	0.050
**Male**												
BMI (kg/m^2^)	16.23	16.80	16.94	0.022	16.14	16.38	16.90	0.336	16.37	16.5	16.56	0.986
Waist (cm)	55.72	56.56	58.10	0.065	54.32	56.29	54.63	0.546	54.78	55.88	54.36	0.511
TG (mmol/L)	0.69	0.78	0.73	0.739	0.74	0.64	0.78	0.620	0.74	0.65	0.79	0.592
HDL-C (mmol/L)	1.27	1.30	1.36	0.037	1.21	1.26	1.28	0.212	1.24	1.26	1.26	0.802
FG (mmol/L)	4.85	4.84	5.02	0.035	4.89	4.79	4.91	0.462	4.97	4.68	4.94	0.054
SBP (mmHg)	91.18	93.13	93.73	0.156	90.44	93.98	93.28	0.025	90.59	93.86	93.14	0.066
DBP (mmHg)	58.83	60.59	61.28	0.026	59.75	61.97	62.09	0.084	59.76	62.22	61.82	0.133
**Female**												
BMI (kg/m^2^)	15.90	16.14	16.43	0.020	15.95	16.30	16.06	0.879	15.96	16.4	15.96	0.879
Waist (cm)	54.32	55.54	55.59	0.274	54.51	53.89	54.43	0.728	53.91	54.21	54.6	0.728
TG (mmol/L)	0.79	0.83	0.93	0.053	0.75	0.87	0.77	0.139	0.79	0.81	0.80	0.139
HDL-C (mmol/L)	1.20	1.26	1.25	0.228	1.14	1.13	1.24	0.011	1.14	1.15	1.22	0.011
FG (mmol/L)	4.68	4.73	4.99	<0.001	4.86	4.75	4.70	0.280	4.89	4.74	4.68	0.280
SBP (mmHg)	89.21	92.09	95.39	<0.001	90.01	89.77	91.40	0.494	89.76	90.89	90.42	0.494
DBP (mmHg)	58.44	60.37	62.00	<0.001	58.46	60.06	60.34	0.272	58.15	61.14	59.51	0.272

TBI—total body iron; BMI—body mass index; TG—triglycerides; HDL-C—high-density lipoprotein cholesterol; FG—fasting glucose; SBP—systolic blood pressure; DBP—diastolic blood pressure.

**Table 3 nutrients-12-01486-t003:** Odds ratios for MetS and its components according to different markers of iron status.

Index		Whole Blood Iron				Ferritin				TBI			
		T1	T2	T3	*p*-Value	T1	T2	T3	*p*-Value	T1	T2	T3	*p*-Value
MetS	Crude	1	4.26 (0.47–7.82)	2.09 (0.18–4.53)	0.536	1	0.85 (0.11–6.42)	0.74 (0.09–5.73)	0.748	1	0.26 (0.03–2.64)	0.51 (0.08–3.25)	0.346
	Adjusted *	1	3.35 (0.36–4.31)	1.96 (0.17–3.21)	0.171	1	1.03 (0.14–7.66)	0.94 (0.15–7.09)	0.936	1	0.26 (0.03–2.77)	0.50 (0.07–3.46)	0.591
Elevated Waist	Crude	1	1.21 (0.78–1.89)	1.22 (0.78–1.91)	0.266	1	1.76 (0.92–3.37)	1.73 (0.89–3.33)	0.362	1	1.25 (0.68–2.28)	1.05 (0.56–1.98)	0.762
	Adjusted *	1	1.11 (0.67–1.86)	1.09 (0.66–1.79)	0.731	1	1.49 (0.76–2.93)	1.40 (0.71–2.74)	0.325	1	1.13 (0.60–2.14)	0.98 (0.61–1.90)	0.958
Hypertension	Crude	1	2.08 (0.62–6.93)	4.36 (1.44–13.21)	0.014	1	0.78 (0.12–4.79)	0.93 (0.15–5.85)	0.43	1	1.56 (0.26–8.82)	0.98 (0.14–7.18)	0.632
	Adjusted *	1	2.21 (0.67–7.34)	4.07 (1.34–12.36)	0.013	1	0.67 (0.11–4.06)	0.64 (0.11–3.92)	0.635	1	1.48 (0.24–8.97)	0.98 (0.13–7.08)	0.988
Elevated TG	Crude	1	1.11 (0.60–2.07)	1.04 (0.56–1.94)	0.137	1	0.78 (0.31–1.95)	0.81 (0.32–2.02)	0.950	1	0.32 (0.11–0.93)	0.76 (0.33–1.72)	0.092
	Adjusted *	1	1.10 (0.59–2.040	1.09 (0.58–2.02)	0.785	1	0.81 (0.32–1.57)	0.86 (0.35–2.10)	0.754	1	0.30 (0.10–0.87)	0.68 (0.29–1.61)	0.032
ReducedHDL-C	Crude	1	0.68 (0.47–1.01)	0.65 (0.44–0.95)	0.053	1	0.97 (0.61–1.54)	0.46 (0.27–0.78)	0.011	1	0.98 (0.61–1.65)	0.61 (0.37–1.00)	0.098
	Adjusted *	1	0.67 (0.46–0.99)	0.63 (0.43–0.93)	0.022	1	0.99 (0.62–1.57)	0.49 (0.29–0.81)	0.006	1	0.94 (0.58–1.51)	0.57 (0.34–0.95)	0.033
Hyperglycemia	Crude	1	1.2 (0.68–2.11)	1.62 (1.03–2.76)	0.012	1	0.88 (0.44–1.78)	0.57 (0.26–1.25)	0.685	1	0.87 (0.55–1.38)	0.65 (0.41–1.05)	0.206
	Adjusted *	1	1.22 (0.70–7.34)	1.74 (1.02–2.95)	0.040	1	0.88 (0.44–1.75)	0.57 (0.26–1.22)	0.148	1	0.42 (0.21–0.87)	0.39 (0.18–0.83)	0.015

TBI—total body iron; BMI—body mass index; TG—triglycerides; HDL-C—high-density lipoprotein cholesterol; FG—fasting glucose; CRP was adjusted in the assessment of ferritin tertiles. *p*-value, *p* for trend. * adjusted for age, gender, BMI, and CRP.

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
