# Peer review of "Association of Iron Storage Markers with Metabolic Syndrome and Its Components in Chinese Rural 6–12 Years Old Children: The 2010–2012 China National Nutrition and Health Survey"

_nutrients, 2020, doi:10.3390/nu12051486_

Round 1

Reviewer 1 Report

This paper describes a secondary analysis of Chinese national data with the objective of assessing the relationship between biomarkers of iron storage and components of metabolic syndrome among rural children aged 6-12 years.

The objectives have merit, but the markers of iron storage have not necessarily been used correctly.  I suggest the authors review the following reference.

Lynch S, Pfeiffer CM, Georgieff MK, Brittenham G, Fairweather-Tait S, Hurrell RF, McArdle HJ, and Raiten DJ. Biomarkers of Nutrition for Development (BOND)—Iron Review. J Nutr 2018;148:1001S–1067S.

Consider the possibility that ferritin—at the levels found in this population—is more a marker of inflammation than an indicator of sub-clinical iron overload.  The purpose of using sTfR to calculate TBI is to assess iron deficiency in the presence of inflammation.

Why were those with low ferritin excluded?

Adjustment for BMI may not be appropriate because it is part of the metabolic syndrome.  Adjustment for CRP does not necessarily remove inflammatory effects of ferritin as is now stated in the discussion.  I suggest a causal model or a directed acyclic graph to clarify hypothesized relationships before multivariable modeling.

In table 1, CRP and ferritin were both high in the small group (n=13) with metabolic syndrome (<=3 components), most likely indicating inflammation.  Combined with the result that TBI was lowest in the small group with metabolic syndrome, it seems most plausible that ferritin is serving as a marker of inflammation and not a marker of sub-clinical iron overload.

Consider the clinical implications of this work.  Would you recommend children at risk for metabolic syndrome consume less iron? 

Author Response

Point 1The objectives have merit, but the markers of iron storage have not necessarily been used correctly. I suggest the authors review the following reference.

Lynch S, Pfeiffer CM, Georgieff MK, Brittenham G, Fairweather-Tait S, Hurrell RF, McArdle HJ, and Raiten DJ. Biomarkers of Nutrition for Development (BOND)—Iron Review. J Nutr 2018;148:1001S–1067S.

Response1: Thank you for the very kind recommendation. We have carefully studied the reference article. It really helps us to know more about the functions of iron indicators, which is very valuable. Some indicators of iron nutritional status associated with other diseases are summarized in table 1. In this study, based on the 2010-2012 blood sample restriction, we only selected the SF, TBI and whole blood iron as the indicator of iron storage. SF is the most commonly used in clinical and public health settings. TBI, which combined sTfR and ferritin, could be comprehensive and sensitive indicators to comprehensively evaluate iron storage. Whole blood sample has been proposed as the best matrix for characterizing iron exposure, because 60–70% of total body iron is present in hemoglobin in circulating erythrocytes. This is also a new attempt in our research. However, we did not cover the other indicators, which can also a valuable marker for iron status, such as transferritin , TSAT.

Table 1 Response of iron status indicators to a depletion of body iron compartments with and without concomitant inflammation and to an overload of body iron compartments

Compartment

Indicator

IDA

ACD

IDA+ACD

Overload

Stored iron

SF

Reduced

Normal to increased

Reduced to normal

Increased

Transport iron

Iron

Reduced

Reduced

Reduced

Increased

Transferritin

Increased

Reduced

Reduced

Reduced

TSAT

Reduced

Reduced

Reduced

Increased

EP

Increased

Increased

Increased

Reduced

sTfR

Increased

Normal

Normal to increased

Normal

Functional iron

Hemogolbin

Reduced

Reduced

Reduced

Normal

Inflammatory response

NA

Normal

Increased

Increased

NA

ACD, anemia of chronic disease; EP, erythrocyte protoporphyrin; IDA, iron deficiency anemia; IDA + ACD, combined iron deficiency anemia and anemia of chronic disease; NA, not applicable; SF, serum ferritin; sTfR, soluble transferrin receptor; TSAT, transferrin saturation.

Point 2:Consider the possibility that ferritin—at the levels found in this population—is more a marker of inflammation than an indicator of sub-clinical iron overload.  The purpose of using sTfR to calculate TBI is to assess iron deficiency in the presence of inflammation. 

Response 2: Thank you for the suggestion. It is really true as you suggested that ferritin may be a marker of inflammation. The threshold of SF recommended by WHO for determining iron excess is 200µg/L for women and 300µg/L for men. But the cut-off value is only for adults and there is no consistency recommendation for children. It is also found that high SF level can increase the risk of MetS when the threshold of iron excess is not reached[1,2]. The purpose of our study is to explore whether SF will increase the risk of MetS or its components in the high tertile of  the normal distribution by using a representative population. In the mean time, considering SF is an acute period protein, which is susceptible to inflammation, we also measured CRP and other indexes, including TBI and whole blood iron, to reflect the iron storage level of the body.

[1] Suárez-Ortegón Milton Fabian,Ensaldo-Carrasco Eduardo,Shi Ting et al. Ferritin, metabolic syndrome and its components: A systematic review and meta-analysis.[J] .Atherosclerosis, 2018, 275: 97-106.

[2] Fang Xuexian,Min Junxia,Wang Fudi,A dose-response association between serum ferritin and metabolic syndrome?[J] .Atherosclerosis, 2018, 279: 130-131.

Point 3:Why were those with low ferritin excluded?

Response 3: The purpose of our study is to analyze whether high tertile compared to low tertile iron markers would increase the risk of MetS components when iron storage is normally distributed. We used logistic regression to detect the association between tertiles of metals and the risk of MetS components. As the first tertile was used as the reference and it also the lowest tertile, we must eliminate the confusion caused by iron deficiency. As a consequence, the ferritin, which was less than <12 µg/L, was excluded. And the elimination of individuals with low ferritin can improve the sensitivity of the whole study and eliminate mixed bias which was caused by iron deficiency.

Point 4:Adjustment for BMI may not be appropriate because it is part of the metabolic syndrome. Adjustment for CRP does not necessarily remove inflammatory effects of ferritin as is now stated in the discussion. I suggest a causal model or a directed acyclic graph to clarify hypothesized relationships before multivariable modeling.

Response 4: Thanks for the valuable suggestion. When we adjusted the regression model, we also considered BMI and CRP as confounding factors. BMI is indeed a common index used to evaluate obesity, but in the definition of metabolic syndrome[1], BMI is not included, but the waist circumference is used to represent abdominal obesity. BMI is a well-known positively marker correlated with iron stores and obesity can also cause inflammation that affects iron storage. In some studies in China[2], Korea[3] and Switzerland[4], BMI was found to influence the ferritin-MetS association and was used as an confounding factor. In this study, we also made the same attempt. We found that the adjustment OR just slightly increased or decreased in MetS components, which had statistically significant with iron markers. SF, being an acute-phase protein, is susceptible to inflammation. Therefore, it is very important and necessary to eliminate the influence of inflammation in the analysis. CRP is one of the indexes of inflammation, which has a very rapid response to inflammation. Although the CRP in our study were all in the normal range 8-80mg/L, the effect of using CRP alone on correcting all inflammation is very limited. Therefore, this is also a drawback of this research. We also added a more detailed description of this limitation.

[1]Magge Sheela N,Goodman Elizabeth,Armstrong Sarah C et al. The Metabolic Syndrome in Children and Adolescents: Shifting the Focus to Cardiometabolic Risk Factor Clustering.[J] .Pediatrics, 2017, 140: undefined.

[2]Sun L, Franco OH, Hu FB, Cai L, Yu Z, Li H, et al. Ferritin concentrations, metabolic syndrome, and type 2 diabetes in middle-aged and elderly chinese. The Journal of clinical endocrinology and metabolism. 2008;93 (12):4690-6.

[3]Cho GJ, Shin JH, Yi KW, Park HT, Kim T, Hur JY, et al. Serum ferritin levels are associated with metabolic syndrome in postmenopausal women but not in premenopausal women. Menopause (New York, NY). 2011;18 (10):1120-4.

[4]Kilani N, Waeber G, Vollenweider P, Marques-Vidal P. Markers of iron metabolism and metabolic syndrome in Swiss adults. Nutrition, metabolism, and cardiovascular diseases : NMCD. 2014;24 (8):e28-9.

Point 5:In table 1, CRP and ferritin were both high in the small group (n=13) with metabolic syndrome (<=3 components), most likely indicating inflammation. Combined with the result that TBI was lowest in the small group with metabolic syndrome, it seems most plausible that ferritin is serving as a marker of inflammation and not a marker of sub-clinical iron overload.

Response 5: Thank you for pointing this out. We are very sorry for our incorrect usage of the punctuation ≥3 into ≤3. In the group with more than 3 MetS components, the value of whole blood iron and ferritin were the highest, whereas the TBI was the lowest. But all the values were still in the normal range based on existing standards. The difference between these indicators is exactly what we hope to explore. We hope to find out the relationship between iron storage and metabolic diseases through the optimization of iron index. We will also conduct validation in future cohort studies.

Point 6:Consider the clinical implications of this work. Would you recommend children at risk for metabolic syndrome consume less iron? 

Response 6: I would like to thank you for raising this important public health proposal. A comprehensive evaluation of the influence of metal on the occurrence and development of diseases requires a full consideration of genes, environment, diet, living habits and other factors. As we know,  there are some reports about the relationship between the intake of total iron, especially heme iron in diet and MS[1,2], but most for adults.

In this study, 3 biological indicators reflecting iron storage were used to evaluate the relationship between body iron status and MetS. However, we did not collect dietary data, especially heme iron data. Although the result indicated that iron storage level in children is associated with the risk of the incidence for MetS components, especially in hyperglycemia and reduced HDL-C. The relationship between the three iron indexes and metabolic syndrome and its components is not completely consistent, which suggests that the underlying mechanism is complex and need further explored. More rigorous and evidence-based epidemiological studies, such as cohort studies and RCT studies, are needed to make recommendations on iron consume in the population. In the future research, our team will combine gene, diet and biological indicators to further study the relationship between iron nutrition and metabolic diseases.
[1]Zhu Zhenni,Wu Fan,Lu Ye et al. Total and Nonheme Dietary Iron Intake Is Associated with Metabolic Syndrome and Its Components in Chinese Men and Women.[J] .Nutrients, 2018, 10: undefined.
[2]Azadbakht Leila,Esmaillzadeh Ahmad,Red meat intake is associated with metabolic syndrome and the plasma C-reactive protein concentration in women.[J] .J. Nutr., 2009, 139: 335-9.

Reviewer 2 Report

The Authors investigated association of iron storage with metabolic syndrome in Chinese children.

The paper lacks of consistency and needs a complete revision.

Some general comments:

  • Subjects: could the Authors explain why they chose to investigate the association in a rural population? And why they selected children aged 6-12 years?
  • Data collection: what about measure of TBI? It's not mentioned in data collection paragraph eventough  it's reported as one of the most significative vairables
  • Results: the classification of patients according to MetS components is misleading: in category 3 < (or >3? please check!) the number of patients (n=13) is too little to draw conclusions
  • line 128, pg 3: I dont' understand the sentence "TBI were all positive (?) in the 4 METs groups"
  • Table 2: HDL-C is higher in T3 for all iron storage markers, isn't it? Could the Authors explain why they conclude that the 3 markers were associate with reduced HDL-C?
  • Many typos are present. Extensive editing of English language is required

Author Response

Point 1Subjects: could the Authors explain why they chose to investigate the association in a rural population? And why they selected children aged 6-12 years?

Response 1: Along with the development of economic level, the living standard and health condition of our residents are constantly improving, and the changes in rural areas are more varied than in city. School age children aged 6-12 are in an important period of physical development.  With the prevalence of obesity in children worldwide, the incidence of metabolic syndrome in children and adolescents has gradually increased. The prevalence of MetS from the 2010-2012 Chinese National Nutrition and Health Survey (CNNS) was 2.4% in children aged 10 to 17 tears old and presented a low age trend. As a consequence, the level of trace elements and the metabolism of glucose and lipid in children in rural China deserve more attention. Thus, we selected the children aged 6-12 in Chinese rural and detected the relationship between iron storage status and the risk of MetS components.

Point 2:Data collection: what about measure of TBI? It's not mentioned in data collection paragraph even though it's reported as one of the most significative vairables

Response 2: Thank you for the kind suggestion. TBI is a indicator of iron storage which is calculated by the logarithm of the ratio of sTfR to SF concentrations and expressed as mg/kg body weight. And we have added a more specific calculation formula of TBI in the data collection section.

Point 3:Results: the classification of patients according to MetS components is misleading: in category 3 < (or >3? please check!) the number of patients (n=13) is too little to draw conclusions

Response 3: We apologize for the error of the classification of MetS components, and we have corrected the text as suggested. Because the population of this study is 6-12 years old children and the prevalence of metabolic syndrome of this population is relatively low. There is now no uniform standard for the diagnosis of children under 10 years old and there are few studies on this part of the population.In this study, we did not determine to study the prevalence of metabolic syndrome in children, but to study the association of metals level on the risk of MetS components.

Point 4:line 128, pg 3: I dont' understand the sentence "TBI were all positive (?) in the 4 METs groups"

Response 4: Thank you for pointing this out. TBI greater than 0 indicates an adequate iron storage in the body[1]. We described the iron status as positive storage with TBI values all beyond zero in the 4 MetS group .

[1]Pfeiffer Christine M,Looker Anne C,Laboratory methodologies for indicators of iron status: strengths, limitations, and analytical challenges.[J] .Am. J. Clin. Nutr., 2017, 106: 1606S-1614S.

Point 5:Table 2: HDL-C is higher in T3 for all iron storage markers, isn't it? Could the Authors explain why they conclude that the 3 markers were associate with reduced HDL-C?

Response 5: Thanks for the good evaluation. There were significantly ascending trends of HDL-C concentrations in all iron storage markers as the tertile raised shown in table 2. When we assessed the odds ratio for MetS components in tertiles of markers of iron storage, we observed that all the iron markers could decrease the prevalence risk of reduced HDL-C. Combining the results of the two tables(table 2 and 3), regardless of whether the iron storage index is a continuous or categorical variable, the correlation of the 3 markers with HDL-C could be seen.

Point 6:Many typos are present. Extensive editing of English language is required

Response 6: We are very sorry for our incorrect usage. We have carefully revised the manuscript according to the comments. We will  re-scrutinized to improve the English by a language polishing service to the requirements of the editorial department.

Reviewer 3 Report

The paper present good piece of work however there are several points which need to be address. 

In conclusion authors wrote that “ Iron storage level within the normal range in children is associated with the risk of MetS components, especially in hyperglycemia and reduced HDL-C”.

and just one statement above “Thehighest tertile of whole blood iron increased risk of the incidence for hyperglycemia(OR=1.74),

 while TBI decreased the risk of 61% for it. What is true?? I understood that Iron storage equals TBI

There are several not adequate citation for example first statement of the discussion “MetS and its components have received widespread attention and discussion since it was raised in 1981 by Sullivan[9].it is not true !!! In my opinion all; the citied papers should be check for their accuracy.

Results Line 122 “the concentration of BMI (16.59±2.89 vs.16.14±2.54kg/m2” BMI cannot be express as concentration.,

In the results section there are no hematological data like RBC, Hb etc

Discussion line 52 “The mainly possible explanations of correlation between iron 52 storage markers and glycolipid

 metabolism ….” Glycolipids are lipids with a carbohydrate attached by a glycosidic (covalent) bond. Did the authors refer to them? 

Discussion line 60 “Insulin damage would affect the concentration ofglucose” what do you mean by this???

Discussion line 72-74 A possible reason for the result of these study could be the range of markers’ concentrations were all far below iron excess levels as the other  research dose[42-44]. The definition of iron surplus of TBI is beyond zero[29]”.

This statements is difficult to understand and they do not explain anything

Author Response

Response to Reviewer 3 Comments

We very appreciate your careful reading of our manuscript and the valuable suggestions. We have carefully considered the comments and revised the manuscript accordingly. The comments can be summarized as follows:

Point 1In conclusion authors wrote that “Iron storage level within the normal range in children is associated with the risk of MetS components, especially in hyperglycemia and reduced HDL-C”. and just one statement above “The highest tertile of whole blood iron increased risk of the incidence for hyperglycemia(OR=1.74), while TBI decreased the risk of 61% for it. What is true?? I understood that Iron storage equals TBI

Response 1: Thank you for the suggestion. The purpose of our study is to analyze whether high tertile compared to low tertile iron storage markers would increase the risk of MetS components when iron storage is normally distributed. Based on the 2010-2012 blood sample restriction, we selected the SF, TBI and whole blood iron as the indicator of iron storage. In this study, we observed that the relationship between the three iron indexes and metabolic syndrome and its components is not completely consistent, which suggests that the underlying mechanism is complex and needs to be further explored.

Point 2:There are several not adequate citation for example first statement of the discussion “MetS and its components have received widespread attention and discussion since it was raised in 1981 by Sullivan[9].it is not true !!! In my opinion all; the citied papers should be check for their accuracy.

Response 2: Thank you so much for pointing this out. We are sorry for the wrong reference. We have carefully checked the references in the article to make sure that each article has been quoted correctly and reasonably.

Point 3Results Line 122 “the concentration of BMI (16.59±2.89 vs.16.14±2.54kg/m2” BMI cannot be express as concentration.,

Response 3: Thank you for the suggestion. We have changed the ‘concentration’ to ‘value’ We will pay more attention to the exact usage of words next time. At the same time, we also invited professional editors to modify and polish the words and grammar of the article (results line 128).

Point 4In the results section there are no hematological data like RBC, Hb etc

Response 4: Thank you for the kind suggestion. This study mainly focuses on the relationship between iron storage markers and metabolic syndrome components. According to the reports[1,2], there is now no evidence that hematology index is related to iron storage and metabolic syndrome. Therefore, the indicators such as RBC and Hb are not included in the analysis. However, we think your suggestion is a good point for exploring the complex underlying mechanism between iron storage index and metabolic syndrome. In future studies, we will include hematology indicators to verify this relationship.

[1]Kang Hee-Taik,Linton John A,Shim Jae-Yong,Serum ferritin level is associated with the prevalence of metabolic syndrome in Korean adults: the 2007-2008 Korean National Health and Nutrition Examination Survey.[J] .Clin. Chim. Acta, 2012, 413: 636-41.

[2]Jehn Megan,Clark Jeanne M,Guallar Eliseo,Serum ferritin and risk of the metabolic syndrome in U.S. adults.[J] .Diabetes Care, 2004, 27: 2422-8.

Point 5Discussion line 52 “The mainly possible explanations of correlation between iron storage markers and glycolipid metabolism ….” Glycolipids are lipids with a carbohydrate attached by a glycosidic (covalent) bond. Did the authors refer to them? 

Response 5: Thank you so much for pointing this out. We are very sorry for the misunderstanding caused by the misuse of vocabulary. Our intention is to refer to glucose metabolism and lipid metabolism. We changed glycolipid metabolism into glucose and lipid metabolism(discussion line 56) .

Point 6Discussion line 60 “Insulin damage would affect the concentration of glucose” what do you mean by this???

Response 6: Thank you for the question. If insulin is damaged, the glucose levels will continue to rise after eat because there's not enough insulin to move the glucose into body's cells. And we changed the “Insulin damage would affect the concentration of glucose” to “Insulin damage would lead to the imbalance of blood glucose regulation ” for better understanding(discussion line 63).

Point 7Discussion line 72-74 A possible reason for the result of these study could be the range of markers’ concentrations were all far below iron excess levels as the other  research dose[42-44]. The definition of iron surplus of TBI is beyond zero[29]”. This statements is difficult to understand and they do not explain anything

Response 7: Thank you for the kind suggestion. This is a possible explanation for the results of this study and the examples cited above. We have changed it to “One possible reason is that, unlike other reports[42–44], the distribution of iron storage indicators in these studies is within the normal range.” for better understanding(discussion line 75).

Reviewer 4 Report

It is an interesting study given the fact that this aspect is not well deined in children. Study design is sound, data analysis is well presented.

Improvement.

  • Limitations need to be discussed in the manuscript
  • The first 3 sentences in the discussion section are more suitable for introduction.

Author Response

Response to Reviewer 4 Comments

We very appreciate your careful reading of our manuscript and the valuable suggestions. We have carefully considered the comments and revised the manuscript accordingly. The comments can be summarized as follows:

Point 1Limitations need to be discussed in the manuscript

Response 1: Thank you for the suggestion. The limitations has been discussed in line 97 to 115. It were explained in combination with some reports. And the limitations mainly included the lack of indicators such as insulin resistance, inflammatory factors and dietary iron and the small sample size and the study population. Meanwhile, at the end of the article, we also plan the future research according to the existing limitations.

Point 2The first 3 sentences in the discussion section are more suitable for introduction.

Response 2: Thank you so much for the kind suggestion. We have moved this part of the content to the introduction, and made adjustments and improvements. Thank you for your patience and careful review of this article. We also invited professional English editors to revise and polish the manuscript, hoping to better convey the content of the article.

Round 2

Reviewer 1 Report

The authors have provided thoughtful and sufficient responses for this reviewer, but as far as I can tell, they have not incorporated any of the points into the new version.  If a reviewer raises questions, it is likely other readers will have the same questions and therefore it indicates a need to provide some clarifying information in the manuscript.  For example, the authors included a response to the reviewer comment asking why those with low ferritin were excluded, but I don't think they have included any clarifying information on that point in the revised version.  If they have, they should alert the reviewer to where in the revised manuscript they have addressed this. The same holds true for all reviewer comments. 

In addition, there are typos throughout this paper that must be addressed before  publication.  For example (and these are only examples, the entire manuscript needs to be carefully reviewed by a professional editor).

  1. Abstract, line 18 "there are few researches." is incorrect grammar and should instead say something like "in children there is limited research."
  2. Abstract, line 30 "lelvel" should be "level"
  3. Discussion, page 8, lines 49-51 "..which still need a further exploration to verify the accident of the result." is completely inappropriate.  Perhaps the authors meant to say "which still needs further exploration to verify the unexpected result."

Author Response

Point 1:The authors have provided thoughtful and sufficient responses for this reviewer, but as far as I can tell, they have not incorporated any of the points into the new version. If a reviewer raises questions, it is likely other readers will have the same questions and therefore it indicates a need to provide some clarifying information in the manuscript.  For example, the authors included a response to the reviewer comment asking why those with low ferritin were excluded, but I don't think they have included any clarifying information on that point in the revised version.  If they have, they should alert the reviewer to where in the revised manuscript they have addressed this. The same holds true for all reviewer comments. 

Response 1: Thank you so much for the kind suggestion. We have added the answers as you suggesting before in the manuscript. We added the explanation of serum ferritin index, the limitation of CRP single index, and the significance of BMI correction in the discussion section. And the references are also supplemented.

Point 2:In addition, there are typos throughout this paper that must be addressed before  publication.  For example (and these are only examples, the entire manuscript needs to be carefully reviewed by a professional editor).

Abstract, line 18 "there are few researches." is incorrect grammar and should instead say something like "in children there is limited research."

Abstract, line 30 "lelvel" should be "level"

Discussion, page 8, lines 49-51 "..which still need a further exploration to verify the accident of the result." is completely inappropriate.  Perhaps the authors meant to say "which still needs further exploration to verify the unexpected result."

Response 2: We are very sorry for our incorrect writing. Thank you so much for pointing this out. we have made correction according to the your comments. And we also invited more professional native speaker to polish and modify the article. Hope to make the article more understandable.

Reviewer 2 Report

I really appreciate the efforts of Authors for answering to my questions.

Unfortunately, I still think that significance of content is too low, mainly due to the small number of patients (n=13) in category 3.

Authors cleary stated that " the population of this study is 6-12 years old children and the prevalence of metabolic syndrome of this population is relatively low" and that "The prevalence of MetS from the 2010-2012 Chinese National Nutrition and Health Survey (CNNS) was 2.4% in children aged 10 to 17 years old".

If their aim is to evaluate the "Association of iron storage markers with metabolic syndrome and its components"   I suggest them to consider extending  their analysis to older patients, too.

Author Response

Point 1:I really appreciate the efforts of Authors for answering to my questions.

Unfortunately, I still think that significance of content is too low, mainly due to the small number of patients (n=13) in category 3.

Authors cleary stated that " the population of this study is 6-12 years old children and the prevalence of metabolic syndrome of this population is relatively low" and that "The prevalence of MetS from the 2010-2012 Chinese National Nutrition and Health Survey (CNNS) was 2.4% in children aged 10 to 17 years old".

If their aim is to evaluate the "Association of iron storage markers with metabolic syndrome and its components"   I suggest them to consider extending their analysis to older patients, too.

Response 1: Thank you for the kind suggestion. In table 1,we divided the subjects into 4 groups according to the number of metabolic syndrome components was see the distribution and trend of the indicators. We also think that the number of people with metabolic syndrome could not be a solid evidence to draw a conclusion. Therefore, this study focuses more on the relationship between iron storage indexes and MetS components. The small sample size of MetS was considered as a limitation shown in the discussion. We really appreciate your advice on the inclusion of subjects. At present, some studies have analyzed the relationship between metabolic syndrome and iron storage index in adolescents, but with the trend of younger age of metabolic syndrome, there is still less research on children especially younger than 10 years old. Our research is a preliminary attempt to study the relationship between MetS and its components and iron storage in the school-age children aged 6-12. In future studies, we will include more age group individuals and further study the relationship between the level of elements and metabolic syndrome and its components in children and adolescents.
